# UrbanCross: Enhancing Satellite Image-Text Retrieval with Cross-Domain Adaptation

## ABSTRACT

Urbanization challenges underscore the necessity for effective satellite image-text retrieval methods to swiftly access specific information enriched with geographic semantics for urban applications. However, existing methods often overlook significant domain gaps across diverse urban landscapes, primarily focusing on enhancing retrieval performance within single domains. To tackle this issue, we present UrbanCross, a new framework for cross-domain satellite image-text retrieval. UrbanCross leverages a high-quality, cross-domain dataset enriched with extensive geo-tags from three countries to highlight domain diversity. It employs the Large Multimodal Model (LMM) for textual refinement and the Segment Anything Model (SAM) for visual augmentation, achieving a fine-grained alignment of images, segments and texts, yielding a 10% improvement in retrieval performance. Additionally, UrbanCross incorporates an adaptive curriculum-based source sampler and a weighted adversarial cross-domain fine-tuning module, progressively enhancing adaptability across various domains. Extensive experiments confirm UrbanCross's superior efficiency in retrieval and adaptation to new urban environments, demonstrating an average performance increase of 15% over its version without domain adaptation mechanisms, effectively bridging the domain gap. Our code is publicly accessible, and the dataset will be made available at https://anonymous.4open.science/r/UrbanCross/.

## CCS CONCEPTS

• **Information systems → Specialized information retrieval**.

## KEYWORDS

Satellite image-text retrieval; Cross-domain adaptation; Multimodal

## 1 INTRODUCTION

Enriched with geographic details, satellite imagery serves as a vital resource for comprehending the functionality of a region, with a variety of applications ranging from poverty assessment [13, 14], crop yield prediction [28, 29], to urban region profiling [10, 38]. *Satellite Image-Text Retrieval* aims to retrieve specific satellite images from an image pool based on text descriptions, and vice versa [3], which has gathered increased attention with global urbanization. The key to the success of such retrieval lies in effectively harmonizing satellite image and textual data within urban complexities [47].

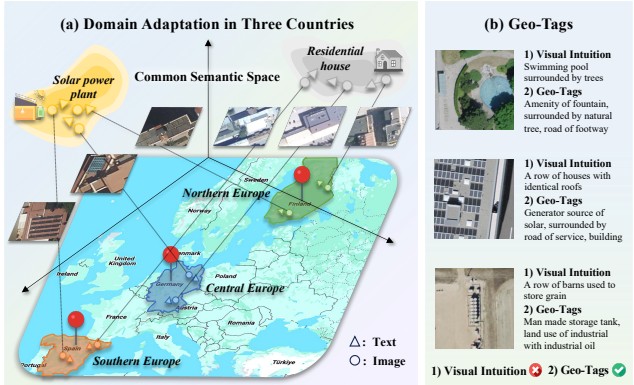

**Figure 1: (a) Domain adaptation for differentiating solar power plants from residential houses across countries. (b) Geo-tags can complement visual intuition.**

Related literature can be categorized into two main paradigms [15, 17, 46]. The first stream, known as *Content-based methods*, is centered on generating precise captions for satellite images via generative models, converting image-text retrieval to a text-to-text task [18, 36]. However, these methods usually encounter detail loss in complex scenes and heavily rely on extensive annotated datasets. In contrast, *Embedding-based methods* aim to leverage pre-trained encoders to align images and texts within a unified semantic space, ensuring better modal interaction for the retrieval task. These approaches explore novel mechanisms, such as self-attention and Contrastive Language-Image Pre-training (CLIP), to enhance representation learning [19, 27, 33] and modality interaction [8, 40].

Though promising, previous approaches mostly assume uniformity in satellite image-text pairs across varied landscapes. This assumption, however, may lead to inferior model performance when dealing with data distribution shifts. Figure 1(a) depicts an example, where geographical differences, such as the varying prevalence of solar power plants between Finland and Spain due to distinct solar exposure levels, may result in the model trained on Finnish data incorrectly classifying Spanish solar plants as residential areas. This underscores the critical need for cross-domain adaptation to ensure semantically equivalent feature alignment across geographies.

In this paper, we identify two key perspectives to enhance satellite image-text retrieval with cross-domain adaptation

• *Data perspective*: Learning domain-invariant features is the prerequisite for cross-domain adaptation. When understanding urban regions, relying solely on visual features may lead to misinterpretation, as illustrated in Figure 1(b). Fortunately, real-world satellite images are often accompanied by other descriptions or metadata, such as geo-tags, which are texts that can generalize well in all countries. By incorporating such auxiliary information, we can effectively identify and complement

original vision representations, thereby enhancing the generalization ability of retrieval models.

- *Model perspective*: Enhancing the model's adaptability to domain shifts requires improving the identification and adjustment capabilities to data variances across domains, through a robust and domain-aware framework. Our approach dynamically adjusts model parameters in response to identified domain-specific features, thereby significantly improving the adaptability and accuracy of the retrieval system.

In response to these challenges, we present a novel framework termed **UrbanCross**, which enhances the embedding-based retrieval paradigm with *cross-domain adaptability*. From the data perspective, we augment data representations with two endeavors. Externally, our model integrates geo-tags with Large Multimodal Model (LMM) to generate visually rich and semantically accurate image captions. Internally, we employ Segment Anything (SAM) to extract fine-grained visual features from the input image itself, eliminating irrelevant elements and aligning these with corresponding text to enhance data quality across visual and language domains.

From the model perspective, we devise two innovative modules: *Adaptive Curriculum-based Source Sampler*, which initially samples source data based on similarity to the target domain, followed by the *Adversarial Cross-Domain Image-Text Fine-tuning Module* for subsequent fine-tuning. This integrated strategy ensures a seamless transition from simpler to complex samples, applying weighting to align with domain-specific traits, thus effectively addressing the challenges posed by diverse data distributions across domains.

Our contributions are summarized as follows:

- **Data Augmentation**: By integrating LMM with geo-tags to enrich textual descriptions and employing SAM for precise visual segmentation, UrbanCross significantly enhances both visual and textual accuracy, ensuring contextual and semantic understanding, resulting in higher-quality data representations.
- **Cross-Domain Adaptation**: We introduce a curriculum-based source sampler and a weighted adversarial fine-tuning module. This integration significantly improves domain adaptation by enhancing the accuracy of multimodal fusion across images, texts, and segmented visual elements.
- **Extensive experiments**: Through extensive comparative and cross-country testing, UrbanCross has achieved a 10% improvement in retrieval performance and a 15% average boost over methods lacking domain adaptation.

## 2 PRELIMINARIES

### 2.1 Formulation

**Definition 1 (Satellite Image)**: A satellite image $I_g$ representing an urban area $g$ can be denoted as in $\mathbb{R}^{H \times W \times 3}$, where $H$ and $W$ are length and width. It includes Ground Sample Distance (GSD) for spatial resolution, geographical coordinates for precise positioning, and geo-tags providing contextual information like location-based labels to aid in object recognition within urban environments.

**Definition 2 (Image Captioning)**: The textual description $T_g$ of a satellite image $I_g$ is generated by LMM, producing captions that integrate visual content and geo-tags. This enriches the understanding of urban features and aids in identifying key objects.

**Problem Statement (Cross-Domain Satellite Image-Text Retrieval)**: Given dataset $D_s = \{(I_{g_i}, T_{g_i})\}_{i=1}^{N_s}$ from a source domain, and $D_t = \{(I_{g'_i}, T_{g'_i})\}_{i=1}^{N_t}$ from a target domain, where $N_s$ and $N_t$ represent the respective lengths of the datasets. The goal is to develop a model $\mathcal{F}$ that maps image-text pairs to vectors within an embedding space that aligns and generalizes across domains. Consequently, the representation vectors $\mathbf{e}_g^I, \mathbf{e}_g^T = \mathcal{F}(I_{g'}, T_{g'})$ can facilitate efficient image-to-text (i2t) and text-to-image (t2i) retrieval tasks.

## 2.2 Related Work

*2.2.1 Satellite Image-Text Retrieval.* Recent developments in satellite image-text retrieval have improved alignment between textual and visual data. Enhanced by neural network architectures like CNNs, RNNs, and Transformers, these advancements foster robust feature extraction and modality interactions [7, 11, 22, 30, 40, 42, 43]. Techniques such as GaLR [43] and KCR [22] have advanced text comprehension by integrating global-local image features and domain-specific knowledge, respectively. Meanwhile, Vision-Language Pre-training (VLP) models like CLIP [27] face challenges in processing diverse urban satellite imagery features. Innovations in high-resolution imaging [33] and multilingual text processing [2] highlight VLP's adaptability, supported by foundational models such as RSGPT [12] and SkyEyeGPT [44]. However, these methods often neglect the data distribution variability across different domains, an issue UrbanCross aims to address.

*2.2.2 Large Multimodal Model and Segment Anything Model.* With the rapid advancements in Large Language Model (LLM), the incorporation of visual information into these models is increasingly prominent. Recent developments, including MiniGPT-4 [4], LLaVA [20], and InstructBLIP [9], have expanded parameters and training data, contributing to the evolution of LMM. Models such as mPLUG-Owl [39], Shikra [6], and KOSMOS-2 [26] have introduced techniques aimed at mitigating hallucinations in LMM. Concurrently, SAM [16], renowned for its robust image segmentation capabilities, finds widespread application across various domains. This includes urban infrastructure analysis [1], UV-SAM for urban segmentation [45], and RSPrompter for satellite image-based instance segmentation [5]. Yet, the potential of these models in satellite image-text retrieval remains largely untapped. Hence, UrbanCross leverages the strengths of LMM and SAM to enrich the image-text dataset, facilitating multimodal alignment and cross-domain adaptation.

*2.2.3 Domain adaptation in Urban Research.* Domain adaptation, a critical subfield of transfer learning, is pivotal in overcoming labeled data scarcity and ensuring model generalization across varied scenarios [25]. In urban areas, the application of domain adaptation is invaluable, leveraging heterogeneous data to markedly enhance model performance in areas such as traffic forecasting [31], environmental monitoring [35], and air quality prediction [34]. Despite its widespread application in urban contexts, domain adaptation strategies remain underexplored in satellite image-text retrieval. UrbanCross addresses this gap by incorporating domain adaptation strategies for satellite imagery and its associated texts, thus enhancing adaptability across global urban landscapes.

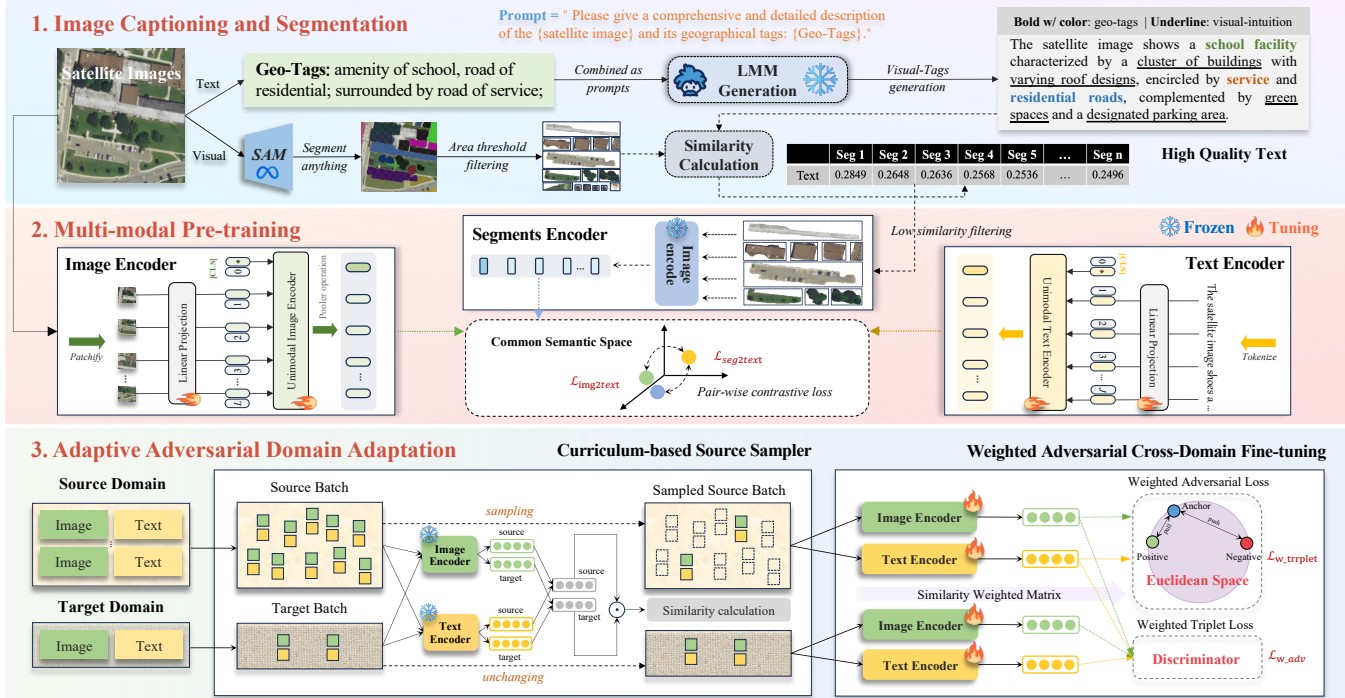

**Figure 2: Overall framework of the proposed UrbanCross.**

## 3 METHODOLOGY

The UrbanCross framework (Figure 2) has three main stages.

- **Image Captioning and Segmentation**: At stage one, LMM, augmented with geo-tags, produces detailed descriptions for satellite images, yielding semantically enriched image-text pairs. Simultaneously, SAM [16] is utilized to segment the image, removing irrelevant areas through area assessment and text similarity comparison, ensuring fine-grained feature extraction for fusion.
- **Multi-Modal Pre-training**: At stage two, images, segments, and texts are encoded independently, then merged into a unified semantic space using pairwise contrastive loss to bring similar ones closer together while pushing dissimilar ones farther apart.
- **Adaptive Adversarial Domain Adaptation**: At stage three, batches from source and target domains are processed by an adaptive curriculum-based sampler. Using the pre-trained encoder, it first transforms image-text pairs into representations, then evaluates domain similarity and progressively excludes source batches, starting with those most similar to the target (easy) to those least similar (hard). Subsequently, refined batches and target domain data undergo adversarial cross-domain fine-tuning, minimizing weighted contrastive and discriminator losses. Unimodal encoders are then fine-tuned for domain alignment.

### 3.1 Image Captioning and Segmentation

*3.1.1 Text Augmentation with LMM and Geo-Tags.* To ensure high-quality textual descriptions for experiments, we employ the InstructBLIP [9] LLM for text generation. However, directly inputting satellite images into the LLM can result in insufficient semantic details. As illustrated in phase 1 of Figure 2, although the model captures general features like "cluster of buildings", "parking areas", and "green spaces", it often omits specific details such as "residential roads" or "school facilities". To address this, geo-tags are integrated to enhance visual accuracy and provide additional contextual information, resulting in n more detailed and precise image captions.

*3.1.2 Image Augmentation with SAM.* In satellite image-text retrieval tasks, precisely matching text details with image features is crucial. However, extraneous elements in satellite images often reduce accuracy. We address this using SAM [16], to isolate key features from images. We initially set an area threshold in SAM's "everything" mode to exclude overly small and non-essential segments, for further improving visual-language alignment.

$$\mathbf{S}_{filtered} = \{\mathbf{s}_i | area(\mathbf{s}_i) > \mathbf{T}_a, \mathbf{s}_i \in \mathbf{S}\}, \quad (1)$$

Here, $\mathbf{S}_{filtered}$, derived by applying an area threshold $\mathbf{T}_a$ to the original segment collection $\mathbf{S}$, excludes smaller segments. CLIP [27] is then used to calculate the similarity between text and segments.

$$score(\mathbf{s}_i, \mathbf{t}) = \langle \boldsymbol{\phi}seg(\mathbf{s}_i), \boldsymbol{\psi}text(\mathbf{t}) \rangle, \quad (2)$$

$$\mathbf{S}_{semantic} = \{\mathbf{s}_i | score(\mathbf{s}_i, \mathbf{t}) > \mathbf{T}_s, \mathbf{s}_i \in \mathbf{S}_{filtered}\}, \quad (3)$$

where $score(\mathbf{s}_i, \mathbf{t})$ measures segment-text similarity, and $\boldsymbol{\phi}seg$ and $\boldsymbol{\psi}text$ are the respective CLIP encoders, segments meeting this criterion are weighted in the final embedding.

$$E_{\mathbf{S}} = \frac{\sum_{i=1}^{n} \mathbf{w}_i \cdot \boldsymbol{\phi}seg(\mathbf{s}_i)}{\sum_{i=1}^{n} \mathbf{w}_i}, \quad (4)$$

where $\mathbf{w}_i = score(\mathbf{s}_i, \mathbf{t})$ serves as the weight for each segment, prioritizing the most relevant ones, ensuring a focused and semantically coherent alignment with textual descriptions.

## 3.2 Multi-modal Pre-training

### 3.2.1 Learning Satellite Image Representations.
Satellite image representations are learned using the CLIP Vision Transformer (ViT-L-14). The image $I_g$ is segmented into 16×16 pixel patches $I_p$, projected to $Ep^I = \mathbf{W}p^I p^\top + bp$ with positional embeddings $\mathbf{E}_{pos}$ added to enhance spatial information. The resulting embeddings $E_e^I = Ep^I + \mathbf{E}_{pos}$ undergo self-attention processing, including multi-head mechanisms (MSA).

$$(\mathbf{Q}^I, \mathbf{K}^I, \mathbf{V}^I)^\top = E_e^I (\mathbf{W}_Q, \mathbf{W}_K, \mathbf{W}_V)^\top, \tag{5}$$

$$E_{(i)}^I = \text{Softmax}(\frac{\mathbf{Q}^I \mathbf{K}^{I^\top}}{\sqrt{d}}) \mathbf{V}^I, \tag{6}$$

$$E_{MSA}^I = \text{Concat}(E_{(1)}^I, E_{(2)}^I, \ldots, E_{(N_h)}^I) \mathbf{W}_O, \tag{7}$$

where $\mathbf{W}_O$ is the learnable output projection matrix, $N_h$ is the number of heads, and Concat indicates concatenation operation. Self-attention dynamically weights the interactions between patches, enhanced by residual connections and layer normalization, producing the latent visual representation:

$$E^I = \text{LayerNorm}(E_e^I + E_{MSA}^I). \tag{8}$$

Building on the dynamic patch representations achieved through self-attention and layer normalization, this approach also incorporates a learnable image [CLS] token to further enhance visual encodings for complex cross-modality tasks.

### 3.2.2 Learning Text Representations.
Text descriptions for satellite images, enhanced by LMM and geo-tags, are processed using a Transformer-Encoder [32]. The text encoder follows a similar MSA mechanism of the visual encoder. The input text sequence is bracketed with [SOS] and [EOS] tokens, and the activation of the highest layer of Transformer at [EOS] token is considered the global representation $\mathbf{E}^T$ of text.

### 3.2.3 Pair-wise Modal Alignment.
To achieve finer feature alignment between visual and language modalities, we enhance retrieval accuracy by creating a shared embedding space that bridges the semantic gap between satellite images ($\mathbf{E}I$), image segments ($\mathbf{E}S$), and text ($\mathbf{E}_T$). This uses pairwise contrastive loss to align images and segments with text, assessing their similarities, respectively:

- **Image-to-Text Contrastive Loss** ($\mathcal{L}_{img2text}$): Reduces disparities between matched image-text pairs while enhancing the distinction for mismatches.
- **Segment-to-Text Contrastive Loss** ($\mathcal{L}_{seg2text}$): Improves the alignment between image segments and text, promoting proximity for related pairs while increasing separation for unrelated ones.

The total objective function, $\mathcal{L} = \mathcal{L}img2text + \mathcal{L}seg2text$, guides the model in tightly aligning images and segments with textual descriptions. By employing margin-driven separation in the embedding space, this contrastive approach enhances the model's ability to differentiate between related and unrelated pairs.

## 3.3 Adaptive Adversarial Domain Adaptation

### 3.3.1 Adaptive Curriculum-based Source Sampler.
The disparity in data distribution between source and target domains hampers domain adaptation. To address this, we introduce the Adaptive

Curriculum-based Source Sampler (ACSS), which progressively integrates more challenging source samples (characterized by their complexity and relevance to the challenges in the target domain) during training. This strategy ensures smooth adaptation to data distribution changes and maintains performance when fine-tuned with limited data. Initially, we concurrently iterate through the source and target datasets with varying data volumes, i.e., batch sizes, selecting a target batch of $n^t$ pairs and a source batch consisting of $n^s = 5 * n^t$ image-text pairs. Features are extracted from these image-text pairs using frozen pre-trained encoders. These are denoted by $\mathbf{E}_P^s = \{\mathbf{E}_{P_i}^s\}_{i=1}^{n^s}$ and $\mathbf{E}_P^t = \{\mathbf{E}_{P_i}^t\}_{j=1}^{n^t}$, where $\mathbf{E}_{P_i}^s$ and $\mathbf{E}_{P_i}^t$ signify the aggregated features of text and images. A similarity matrix $\mathbf{W}_1 \in \mathbb{R}^{n^t \times n^s}$ is then computed to assess the pairwise similarities between the target and source batches, quantifying the degree of alignment between features.

$$\mathbf{W}_1(j, i) = \frac{\mathbf{E}_{P_j^t} \cdot \mathbf{E}_{P_i^s}}{||\mathbf{E}_{P_j^t}|| ||\mathbf{E}_{P_i^s}||}. \tag{9}$$

Based on $\mathbf{W}_1$, we extract a new source subset by selecting the top $K\%$ most similar pairs for each target. The curriculum learning strategy incrementally increases $K$ from 0 by 20% across five epochs, progressing from simpler to more complex samples. Specifically, the range initially covers the first 20%, and then expanding in subsequent epochs to include the next 20% increments, i.e., from 20% to 40%, then from 40% to 60%, and so on. This method ensures progressive exposure from simpler to more complex samples, enhancing adaptation without introducing additional overhead.

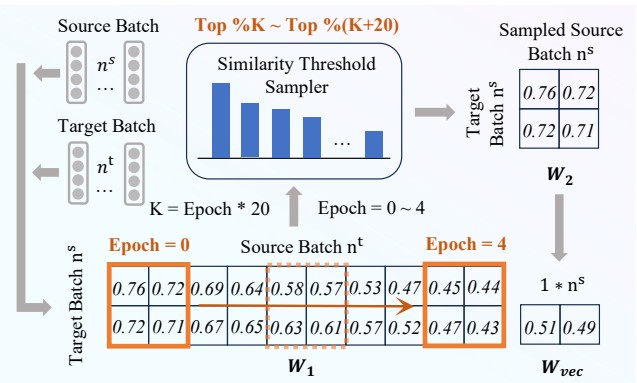

**Figure 3: The similarity matrix between source/target image-text pairs gradually transforms into the weight vector.**

Additionally, $\mathbf{W}_2 \in \mathbb{R}^{n \times n}$, derived from $\mathbf{W}_1$, reflects evolving similarity between selected source and target batches. Continuously updated, these matrices guide the choice of increasingly challenging source batches. Consequently, the ACSS-selected matrix supports weighted adversarial learning, enhancing domain adaptation.

### 3.3.2 Weighted Adversarial Cross-Domain Fine-tuning.
Weighted cross-modal adversarial learning enhances performance by selectively emphasizing source samples similar to target data and reducing the impact of dissimilar ones. As depicted in Figure 2, shared visual and textual encoders process features from both source and target image-text pairs, denoted as $\tilde{\mathbf{E}}_T^s = \{(\tilde{\mathbf{E}}_{\tilde{T}_i^s})\}_{i=1}^n$, $\tilde{\mathbf{E}}_I^s = \{(\tilde{\mathbf{E}}_{\tilde{I}_i^s})\}_{i=1}^n$,

$\tilde{\mathbf{E}}_T^t = \{(\tilde{\mathbf{E}}_{\tilde{T}_j^t})\}_{j=1}^n$ and $\tilde{\mathbf{E}}_I^t = \{(\tilde{\mathbf{E}}_{\tilde{I}_j^t})\}_{j=1}^n$ for batch size $n$. Based on ACSS-selected matrix $W_2$, we get a weight vector $\mathbf{W}_{vec}$ through sum up and normalize across the target dimension, assigns weights to each source sample in training, where $\mathbf{S}$ represents vector elements equal to the sum of each row, specifically, $\mathbf{S}(i) = \sum_{j=1}^n \mathbf{W}_2(i, j)$.

$$
\begin{aligned}
\mathbf{W}_{vec}(i) &\leftarrow \frac{\mathbf{S}(i) - \min(\mathbf{S})}{\max(\mathbf{S}) - \min(\mathbf{S})} \\
\mathbf{W}_{vec}(i) &\leftarrow \frac{n \cdot \mathbf{W}_{vec}(i)}{\sum_{i=1}^n \mathbf{W}_{vec}(i)}
\end{aligned}
\tag{10}
$$

The training employs an objective function $\mathcal{L} = \mathcal{L}_{w_{triplet}^s} + \beta\mathcal{L}_{w_{adv}}$, where $\beta$ modulates the balance between weighted triplet and adversarial losses, with the superscript $s$ indicating that these losses are specific to the source domain. The weighted triplet loss aims to bring positive pairs closer and separate negative pairs, adjusting source data weights during its computation to bridge the gap between source and target domains:

$$
\begin{aligned}
\mathcal{L}_{\mathbf{w}-triplet}^s = \ &\mathbf{W}_{vec}[d(\tilde{\mathbf{E}}_{\tilde{T}_a^s}, \tilde{\mathbf{E}}_{\tilde{I}_p^s}) - d(\tilde{\mathbf{E}}_{\tilde{T}_a^s}, \tilde{\mathbf{E}}_{\tilde{I}_n^s}) + \alpha]_+ \\
&+ \mathbf{W}_{vec}[d(\tilde{\mathbf{E}}_{\tilde{I}_a^s}, \tilde{\mathbf{E}}_{\tilde{T}_p^s}) - d(\tilde{\mathbf{E}}_{\tilde{I}_a^s}, \tilde{\mathbf{E}}_{\tilde{T}_n^s}) + \alpha]_+,
\end{aligned}
\tag{11}
$$

Here, $d(\cdot, \cdot)$ denotes the cosine similarity distance, with $a$, $p$, and $n$ representing anchor, positive, and negative samples, respectively. The term $\alpha$ refers to the margin. A domain discriminator $D$, implemented by a three-layer MLP, is utilized to align source and target domain distributions. During training, $D$ minimizes binary cross-entropy loss between its predictions and ground truth labels, indicating real or generated data. With source $(\tilde{\mathbf{E}}_T^s, \tilde{\mathbf{E}}_I^s)$ and target $(\tilde{\mathbf{E}}_T^t, \tilde{\mathbf{E}}_I^t)$ features as input, $D$ is adversarially trained with the encoder to predict domain labels. The encoders for text and images are fine-tuned to produce image-text features indistinguishable by $D$, which seeks to maximize the probability of correct predictions.

$$
\mathcal{L}_{w\_adv} = \mathbf{W}_{vec} \log D(\tilde{\mathbf{E}}_T^s, \tilde{\mathbf{E}}_I^s) + \mathbf{W}_{vec} \log(1 - D(\tilde{\mathbf{E}}_T^t, \tilde{\mathbf{E}}_I^t)).
\tag{12}
$$

## 4 EXPERIMENTS

In this section, we conduct extensive experiments to investigate the following Research Questions (RQ):

- **RQ1**: Can UrbanCross outperform previous methods without domain adaptation? How does the inclusion of segmented images affect retrieval effectiveness?
- **RQ2**: How effective is UrbanCross in terms of cross-domain adaptation? How does each component (e.g., source sampler, curriculum learning adjustment, adversarial training) contribute to UrbanCross's domain adaptation capability?
- **RQ3**: How does each key hyperparameter (e.g., segmentation number, batch size, learning rate) affect UrbanCross?
- **RQ4**: How does UrbanCross's domain adaptation capability affect satellite image-text retrieval qualitatively?

## 4.1 Experimental Setup

*4.1.1 Datasets.* To assess the efficacy of UrbanCross in satellite image-text retrieval, experiments were performed utilizing the RSICD [21] and RSITMD [41] datasets, which comprise 10,921 and 4,743 images, respectively. Our domain adaptation studies utilized the Skyscript benchmark dataset [37], which features 5.2 million

image-text pairs globally. To improve UrbanCross's adaptability across diverse urban environments, we selected high-resolution images from Spain, Germany, and Finland (GSD ≤ 0.5 m/pixel), resulting in the creation of UC-Spain, UC-Finland, and UC-Germany datasets. These datasets contain 46,041 pairs of 1,621 types of geo-tags from various Spanish regions (ranging from the capital to smaller towns), 165,128 pairs of 3,033 types of geo-tags from Berlin, Germany, and 58,783 pairs of 5,826 types of geo-tags covering Finland, demonstrating a commitment to geographic diversity and data quality. Figure 4 displays data statistics and visualizations.

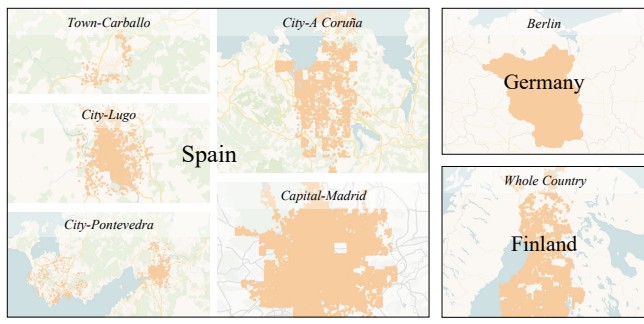

| Country | GSD | Coverd Areas | Image-Text # | Geo-Tags Category # |
|---------|-----|--------------|--------------|---------------------|
| Spain | 0.1 | Madrid, Pontevedra, Lugo, A Coruña, Carballo | 46,041 | 1,621 |
| Germany | 0.2 | Berlin | 165,217 | 5,826 |
| Finland | 0.5 | Whole Country | 59,781 | 3,033 |

**Figure 4: UrbanCross dataset statistics.**

*4.1.2 Baselines.* We compare UrbanCross with the following baselines in satellite image-text retrieval:

- **LW-MCR** [42]: A lightweight algorithm is optimized for multi-scale information retrieval, leveraging insights from latent knowledge within various layers to enhance performance.
- **AMFMN** [40]: It utilizes a multiscale structure and a semantic alignment mechanism to filter out redundant information and align high-level image features with textual data.
- **GaLR** [43]: It incorporates global and local information through a multi-level dynamic fusion module, integrating features across different levels, ensuring a more comprehensive representation.
- **SWAN** [24]: It enhances scene perception and minimizes semantic confusion by introducing a scene-aware aggregation network with a new metric for scene-level retrieval performance.
- **PIR** [23]: It employs a Prior Instruction Representation framework with progressive attention encoders to improve feature representation and long-range dependency modeling, reducing semantic confusion with a cluster-wise attribution loss.
- **RemoteCLIP** [19]: The first vision-language foundational model designed specifically for satellite imagery, aiming for robust feature learning and accurate textual embedding alignment.

*4.1.3 Metrics and Implementation.* Essential metrics are employed to evaluate the performance of satellite image-text retrieval and the effectiveness of domain adaptation. The **R@K** metric, crucial for assessing the accuracy of satellite image-text retrieval, determines whether the correct answer is among the top K results. Mean Recall

**Table 1: Performance comparison of UrbanCross and the state-of-the-art methods on RSICD and RSITMD datasets using R@1, R@5, R@10, and Mean Recall metrics. Includes UrbanCross performance across datasets from Spain, Finland, and Germany, and effectiveness analysis of image segments from the Segment Anything Model via ablation experiments. Optimal and suboptimal performances are indicated by bold and underlined text, respectively.**

| Testing Dataset | Training Dataset | Dataset Size # | Year | Method | Image Backbone | Text Backbone | Image to Text | | | Text to Image | | | Mean Recall |
|---|---|---|---|---|---|---|---|---|---|---|---|---|---|
| | | | | | | | R@1 | R@5 | R@10 | R@1 | R@5 | R@10 | |
| RSICD | RSICD | 10,921 | 2021 | LW-MCR | SqueezeNet | / | 3.54 | 11.89 | 18.78 | 4.23 | 16.67 | 28.01 | 13.85 |
| | | | 2022 | AMFMN | ResNet-18 | GRU | 5.33 | 13.92 | 20.23 | 4.01 | 16.12 | 28.32 | 14.66 |
| | | | 2022 | GaLR | ResNet-18 | GRU | 6.47 | 18.85 | 29.16 | 4.65 | 18.72 | 30.92 | 18.13 |
| | | | 2023 | SWAN | ResNet-50 | GRU | 7.28 | 19.97 | 29.08 | 5.54 | 21.56 | 37.21 | 20.11 |
| | | | 2023 | PIR | Swin-T + ResNet-50 | BERT | 9.97 | 27.32 | 39.33 | 7.01 | 24.53 | 38.88 | 24.51 |
| | | | 2024 | UrbanCross-MMA w/o SEG | ViT-L-14 | Transformer | 17.52 | **38.49** | **51.86** | 14.52 | **40.89** | **57.67** | **36.83** |
| | | | 2024 | UrbanCross-MMA | ViT-L-14 | Transformer | 18.19 (-3.8%) | 37.09 (-3.6%) | 51.46 (-0.8%) | 14.14 (-2.6%) | 39.85 (-2.6%) | 56.43 (-2.2%) | 36.19 (-1.7%) |
| | RET-3 | 165,745 | 2023 | RemoteCLIP | ResNet-50 | Transformer | 12.90 | 32.02 | 44.46 | 10.59 | 33.25 | 48.93 | 30.36 |
| | DET-10 | | 2023 | RemoteCLIP | ViT-B-32 | Transformer | 17.14 | 37.92 | 51.78 | 13.77 | 37.10 | 54.15 | 35.31 |
| | SEG-4 | | 2023 | RemoteCLIP | ViT-L-14 | Transformer | **18.42** | 37.48 | 51.12 | **14.69** | 40.03 | 56.62 | 36.39 |
| RSITMD | RSITMD | 4,743 | 2021 | LW-MCR | SqueezeNet | / | 10.11 | 25.58 | 39.87 | 7.52 | 30.23 | 50.78 | 27.35 |
| | | | 2022 | AMFMN | ResNet-18 | GRU | 10.74 | 23.78 | 40.05 | 10.30 | 34.49 | 54.67 | 29.01 |
| | | | 2022 | GaLR | ResNet-18 | GRU | 11.76 | 29.28 | 41.45 | 9.68 | 35.88 | 54.03 | 30.35 |
| | | | 2023 | SWAN | ResNet-50 | GRU | 13.29 | 30.69 | 45.90 | 10.12 | 39.30 | 60.43 | 33.29 |
| | | | 2023 | PIR | Swin-T + ResNet-50 | BERT | 18.81 | 41.15 | 53.10 | 13.67 | 42.35 | 62.88 | 38.66 |
| | | | 2024 | UrbanCross-MMA w/o SEG | ViT-L-14 | Transformer | **27.98** | **51.68** | 65.56 | 23.66 | 58.44 | 73.78 | **50.18** |
| | | | 2024 | UrbanCross-MMA | ViT-L-14 | Transformer | 27.78 (-0.7%) | 51.22 (-0.89%) | **66.44 (+1.43%)** | 23.73 (+0.3%) | 57.11 (-2.3%) | 71.32 (-3.3%) | 49.60 (-1.2%) |
| | RET-3 | 165,745 | 2023 | RemoteCLIP | ResNet-50 | Transformer | 22.79 | 47.79 | 61.95 | 19.42 | 51.64 | 70.58 | 45.70 |
| | DET-10 | | 2023 | RemoteCLIP | ViT-B-32 | Transformer | 27.65 | 50.88 | 65.93 | 21.99 | 56.11 | 73.27 | 49.31 |
| | SEG-4 | | 2023 | RemoteCLIP | ViT-L-14 | Transformer | 27.88 | 51.55 | 63.27 | 23.63 | 59.42 | **74.82** | 50.10 |
| UC-Spain | UC-Spain | 46,041 | 2024 | UrbanCross-MMA w/o SEG | ViT-L-14 | Transformer | 6.72 | 19.73 | 28.63 | 7.62 | 21.61 | 30.87 | 19.20 |
| | | | 2024 | UrbanCross-MMA | ViT-L-14 | Transformer | 7.81 (+16.2%) | 22.12 (+12.1%) | 31.98 (+11.7%) | 8.44 (+10.8%) | 23.79 (+10.1%) | 33.6 (+8.8%) | 21.29 (+10.9%) |
| UC-Finland | UC-Finland | 59,781 | 2024 | UrbanCross-MMA w/o SEG | ViT-L-14 | Transformer | 8.98 | 25.15 | 35.49 | 8.78 | 24.69 | 35.75 | 23.14 |
| | | | 2024 | UrbanCross-MMA | ViT-L-14 | Transformer | 10.4 (+15.8%) | 26.95 (+7.2%) | 37.29 (+5.1%) | 10.32 (+17.5%) | 28.08 (+13.7%) | 38.71 (+8.3%) | 25.29 (+9.3%) |
| UC-Germany | UC-Germany | 165,217 | 2024 | UrbanCross-MMA w/o SEG | ViT-L-14 | Transformer | 9.33 | 26.38 | 37.01 | 9.11 | 25.37 | 36.52 | 23.95 |
| | | | 2024 | UrbanCross-MMA | ViT-L-14 | Transformer | 10.62 (+13.8%) | 27.86 (+5.6%) | 39.46 (+6.62%) | 10.98 (+20.53%) | 29.28 (+15.41%) | 39.54 (+8.3%) | 26.29 (+9.8%) |

(**MeanR**), providing a comprehensive overview of performance, computes the average of R@1, R@5, and R@10 for both image-to-text and text-to-image satellite retrieval tasks.

Furthermore, to evaluate domain adaptation, UrbanCross is pretrained on data from one country and tested in another, assessing the model's capacity to generalize to different urban environments. Comparing **MeanR** performance—direct model transfer versus fine-tuning with a subset of target domain data—highlights significant enhancements in global urban analysis effectiveness.

Our framework is trained on NVIDIA A800 GPUs utilizing the Adam Optimizer, with hyperparameter adjustments based on each epoch's performance on the validation set. During the multimodal pretraining stage, the initial learning rate is established at 1e-6, implementing a dynamic reduction strategy with a weight decay of 0.3 every 10 epochs. Batch sizes are set at 40, spanning 15 epochs, with each image keeping segments num as 6. Threshold $T_a$ and $T_s$ are set to 0.2. In the domain adaptation fine-tuning phase, the learning rate is set at 1e-7, spanning 5 epochs. The ratio, defined by the target batch size of 16 to the source batch size of 80, is 0.2. $\beta$ of the loss function is set to 1. For the pretraining dataset, we split train:val:test=7:1:2, while for the fine-tuning dataset, we split train:val:test=2:1:7, to facilitate domain-adaptation fine-tuning under limited data scenarios.

## 4.2 RQ1: Retrieval Performance Evaluation

An empirical evaluation was performed to assess the performance of various models on satellite image-text retrieval tasks, utilizing the RSICD and RSITMD datasets. This evaluation aimed to benchmark the effectiveness of the UrbanCross model against conventional methodologies. Additional experiments were conducted on the UC-Spain, UC-Finland, and UC-Germany datasets to establish benchmarks for future research. Importantly, this comparison excluded considerations of domain adaptation. The UrbanCross variant deployed in these assessments was UrbanCross-MMA (Multi-Modal Alignment), which excluded the domain adaptation stage. Furthermore, an ablation study was conducted to explore the impact of image segmentation on enhancing retrieval accuracy. The detailed results of these experiments are presented in Table 1.

- UrbanCross exhibits improved retrieval accuracy, surpassing the recent baseline, PIR, with mean recall improvements of 50.3% on RSICD dataset and 29.8% on RSITMD dataset, under equivalent training and testing conditions. Besides, it also demonstrates a modest enhancement over RemoteCLIP, which is trained on a larger dataset, highlighting its effective learning capabilities.
- The impact of SEG varies across datasets, depending on the quality of the data. For the RSICD and RSITMD datasets, adding segments for alignment decreases mean recall by 1.7% and 1.2%, respectively. This reduction is primarily attributed to the insufficient semantic information in the text, which hinders alignment with the fine-grained segment features, leading to broader contextual matching and, consequently, diminished returns.
- Conversely, the SEG-inclusive model excels in UC datasets with rich annotations, showing mean recall improvements of 10.9% in UC-Spain, 9.3% in UC-Finland, and 9.8% in UC-Germany. These gains highlight SEG's importance in datasets requiring fine-grained visual-textual matching. Our method, characterized by

**Table 2: UrbanCross Domain Adaptation and Ablation Study: Assesses Mean Recall effects when omitting Source Sampler (SS), Curriculum Learning Adjustment (CL), and Adversarial Training (AT) in cross-domain retrieval. Underlined text denotes no domain adaptation; bold text denotes full domain adaptation.**

| Domain Adaptation | | Method | Image to Text | | | Text to Image | | | Mean |
|---|---|---|---|---|---|---|---|---|---|
| Source | Target | | R@1 | R@5 | R@10 | R@1 | R@5 | R@10 | Recall |
| Finland
59,781 # | Spain
46,041 # | UrbanCross-MMA | 0.60 | 2.48 | 3.87 | 0.71 | 2.67 | 4.28 | 2.44 |
| | | UrbanCross (w/o SS) | 0.64 (+6.7%) | 2.55 (+2.8%) | 3.99 (+3.1%) | 0.74 (+4.2%) | 2.80 (+4.9%) | 4.50 (+5.1%) | 2.54 (+4.2%) |
| | | UrbanCross (w/o CL) | 0.66 (+10%) | 2.62 (+5.6%) | 4.08 (+5.4%) | 0.76 (+7.0%) | 2.88 (+7.9%) | 4.64 (+8.4%) | 2.61 (+7.0%) |
| | | UrbanCross (w/o AT) | 0.68 (+13.3%) | 2.69 (+8.5%) | 4.17 (+7.8%) | 0.78 (+9.9%) | 2.95 (+10.5%) | 4.78 (+11.7%) | 2.68 (+9.9%) |
| | | **UrbanCross** | **0.71 (+18.3%)** | **2.82 (+13.7%)** | **4.35 (+13.7%)** | **0.82 (+15.5%)** | **3.08 (+15.4%)** | **5.02 (+17.3%)** | **2.80 (+15.0%)** |
| Germany
165,217 # | Spain
46,041 # | UrbanCross-MMA | 1.50 | 5.30 | 8.40 | 1.90 | 5.60 | 8.80 | 5.25 |
| | | UrbanCross (w/o SS) | 1.55 (+3.3%) | 5.45 (+2.8%) | 8.65 (+3.0%) | 1.96 (+3.2%) | 5.78 (+3.2%) | 9.06 (+3.0%) | 5.41 (+3.0%) |
| | | UrbanCross (w/o CL) | 1.60 (+6.7%) | 5.60 (+5.7%) | 8.90 (+6.0%) | 2.02 (+6.3%) | 5.96 (+6.4%) | 9.32 (+6.1%) | 5.73 (+9.1%) |
| | | UrbanCross (w/o AT) | 1.65 (+10.0%) | 5.75 (+8.5%) | 9.15 (+8.9%) | 2.08 (+9.5%) | 6.14 (+9.6%) | 9.58 (+8.9%) | 5.89 (+12.4%) |
| | | **UrbanCross** | **1.72 (+14.7%)** | **6.10 (+15.1%)** | **9.66 (+15.0%)** | **2.19 (+15.3%)** | **6.46 (+15.4%)** | **10.16 (+15.5%)** | **6.05 (+15.2%)** |
| Germany
165,217 # | Finland
59,781 # | UrbanCross-MMA | 1.10 | 3.80 | 6.30 | 1.50 | 4.70 | 7.80 | 4.20 |
| | | UrbanCross (w/o SS) | 1.14 (+3.6%) | 3.93 (+3.4%) | 6.52 (+3.5%) | 1.55 (+3.3%) | 4.86 (+3.4%) | 8.07 (+3.5%) | 4.34 (+3.3%) |
| | | UrbanCross (w/o CL) | 1.18 (+7.3%) | 4.06 (+6.8%) | 6.74 (+7.0%) | 1.60 (+6.7%) | 5.02 (+6.8%) | 8.34 (+6.9%) | 4.49 (+6.9%) |
| | | UrbanCross (w/o AT) | 1.22 (+10.9%) | 4.19 (+10.3%) | 6.96 (+10.5%) | 1.65 (+10.0%) | 5.18 (+10.2%) | 8.61 (+10.4%) | 4.63 (+10.2%) |
| | | **UrbanCross** | **1.27 (+15.5%)** | **4.39 (+15.5%)** | **7.28 (+15.6%)** | **1.73 (+15.3%)** | **5.42 (+15.3%)** | **9.00 (+15.4%)** | **4.85 (+15.5%)** |

high-quality texts, geo-tags, and LMM integration, effectively enhances image-text alignment, especially in datasets with thorough semantics and contextual information, as shown by the improved recall figures for UrbanCross with SEG, demonstrating how image segmentation significantly improves the precision and relevance of data retrieval in environments with detailed visual and textual data.

## 4.3 RQ2: Cross-Domain Adaptability Evaluation

In this study section, we conducted experiments to evaluate the domain adaptation effectiveness of UrbanCross across various countries. The analysis leverages three main transfer scenarios: Finland to Spain, Germany to Spain, and Germany to Finland. Each scenario involves comparative ablation studies to understand the impact of each adaptation component: the Source Sampler (SS), Curriculum Learning Adjustment (CL), and Adversarial Training (AT). Detailed comparison data is shown in table 2. These findings demonstrate:

- **Overall Transferability Improvement**: UrbanCross demonstrates significant adaptability improvements in all scenarios when fully configured with SS, CL, and AT. For instance, complete configurations yield increases in mean recall of 15.0% for Finland to Spain, 15.2% for Germany to Spain, and 15.5% for Germany to Finland. These improvements underscore the model's robustness in effectively managing domain shifts and enhancing retrieval accuracy across varied urban datasets.
- **Effectiveness of Source Sampler (SS)**: The ablation of SS demonstrates the most significant decrease in performance enhancement compared to removing other components, emphasizing its critical role. As we can see, the presence of SS ensures that the model adapts to changes in data distribution from the source domain to the target domain. Without such adaptation,

the data gap between the source and target domains would lead to a decline in performance. Specifically, gains after removing SS are +4.2% for Finland to Spain, +3.0% for Germany to Spain, and +3.3% for Germany to Finland. These modest gains confirm the effectiveness of SS in bridging the data distribution gap between source and target domains.
- **Effectiveness of Curriculum Learning Adjustment (CL)**: Removing CL results in higher performance gains than removing AT but lower than removing SS, placing it in the middle of the adaptation hierarchy. Specifically, the performance gains after removing CL are +7.0% for Finland to Spain, +9.1% for Germany to Spain, and +6.9% for Germany to Finland. This indicates that curriculum learning strategy allows the source sampler to dynamically adjust filter threshold, thus enhancing the learning difficulty. This guides the model to focus on more challenging examples in the later stages of training, thereby significantly improving domain adaptation performance.
- **Effectiveness of Adversarial Training (AT)**: The exclusion of AT, while still maintaining SS and CL, shows the least impact on performance compared to the other components. The gains are +9.9% for Finland to Spain, +12.4% for Germany to Spain, and +10.2% for Germany to Finland. Although AT contributes to model robustness and generalization, its impact on performance enhancement is less significant than SS and CL, making it the least influential component in the adaptation process.

## 4.4 RQ3: Hyperparameter Study

The optimization process of UrbanCross's hyperparameters is meticulously designed to enhance the mean recall, a critical performance metric. This process is comprehensively visualized in Figure 5, which delineates the adjustments and their impacts during both the pre-training and fine-tuning stages.

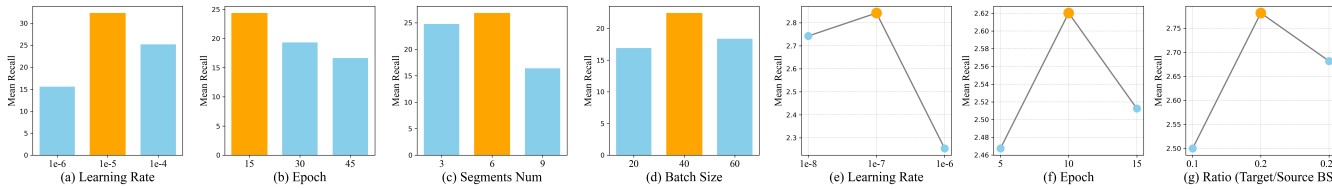

**Figure 5: Impact of hyperparameters on Mean Recall during UrbanCross Pre-training (a-d) and Fine-tuning (e-g).**

- **Pre-training Stage**: Initial analysis examines the learning rate's influence on model performance. A learning rate of 1e-5 optimizes mean recall, offering a balance between convergence speed and stability. Increasing the learning rate to 1e-4 results in diminished recall, indicative of surpassing optimal parameters. Additionally, extending pre-training beyond 30 epochs does not improve recall, signaling a dataset learning saturation point. The number of segments significantly influences performance; six segments are optimal for capturing pertinent patterns without overfitting. A batch size of 40 is ideal, striking a balance between computational efficiency and generalization capacity.

- **Fine-tuning Stage**: The fine-tuning stage begins by examining the learning rate's impact on model performance during domain adaptation An optimal rate of 1e-7 refines parameters with minimal deviation. Extending fine-tuning beyond 10 epochs leads to overfitting, highlighted by decreased recall, emphasizing the need for moderation in epoch selection. A target-to-source batch size ratio of 0.2 (80/16) optimizes mean recall, balancing domain-specific learning with knowledge retention.

Through systematic experimentation during both the pre-training and fine-tuning stages, we identified optimal settings that maximize mean recall while ensuring model stability.

## 4.5 RQ4: Qualitative Analysis

To demonstrate our method's cross-domain efficacy, we provide two illustrative examples representing distinct settings: Finland to Spain and Germany to Finland, as depicted in Figure 6.

For the image query of an industrial roof with solar panels, using the pre-trained model from Finland directly in Spain, without domain adaptation, resulted in inferior outcomes and failed to accurately capture the solar panel concept. This discrepancy can be attributed to the climatic differences between Finland and Spain. Finland experiences limited annual sunlight and extended periods of darkness, particularly in winter, while Spain enjoys abundant sunshine and widespread solar energy adoption, resulting in frequent solar panel depictions in image-text pairs. By fine-tuning the model through domain adaptation with a small dataset from Spain, we achieved accurate textual descriptions. An analysis of the top four results reveals a consensus regarding the solar panel concept.

In the second example, assessing the model's ability to interpret test query images underscores features characteristic of Finland, such as extensive snow cover and a vibrant winter sports scene featuring skiing, ice hockey, and ice skating. Conversely, the absence of comparable samples in Germany results in pre-fine-tuning images that emphasize superficial attributes like snow and whiteness, potentially yielding irrelevant results unrelated to ice sports. These

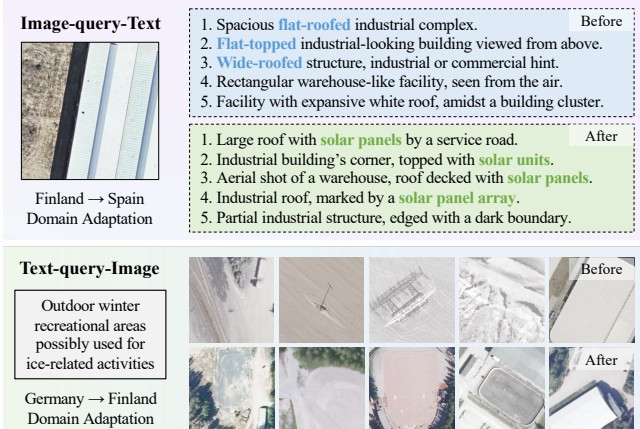

**Figure 6: UrbanCross domain adaptation: before and after comparison. Top 5 retrieval results for image-to-text retrieval from Finland to Spain (above) and text-to-image retrieval from Germany to Finland (below).**

images might portray snow-covered locations with no specific association with ice sports activities. Through fine-tuning, the model significantly enhances its precision in associating search text with locations related to ice sports.

## 5 CONCLUSION AND FUTURE WORK

This paper introduced UrbanCross, a novel framework designed for cross-domain satellite image-text retrieval, achieving enhancements at both data and model levels. At the data level, it integrates geo-tags and an LMM to enrich textual semantics and diversity, and employs SAM to maintain highly relevant visual features, thus ensuring improved alignment with text. At the model level, it incorporates fine-grained feature fusion for enhanced modal alignment and introduces a novel domain adaptation module that combines a Curriculum-based Source Sampler with Weighted Adversarial Cross-Domain Fine-tuning to effectively address the domain gap across various countries. Experimental results demonstrate that UrbanCross surpasses baseline models in retrieval performance and exhibits remarkable adaptability to diverse urban landscapes.

We aspire that this work will inspire future research in cross-domain satellite image-text retrieval frameworks on the following aspects: 1) Designing more precise curriculum learning strategy; 2) Exploring zero-shot cross-domain satellite image-text retrieval; 3) Empowering knowledge update via external knowledge base.

UrbanCross: Enhancing Satellite Image-Text Retrieval with Cross-Domain Adaptation

ACM MM, 2024, Melbourne, Australia

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
