# OpenReview forum: "UrbanCross: Enhancing Satellite Image-Text Retrieval with Cross-Domain Adaptation"
_acmmm.org/ACMMM/2024/Conference — MM2024 Poster_

### Official Review · Reviewer_pzKN · 2024-05-17

**Rating:** 3
**Confidence:** 4

**Summary:**

This manuscript introduces UrbanCross, a novel framework addressing the need for efficient satellite image-text retrieval methods in urban applications. Unlike existing methods focusing on single domains, UrbanCross emphasizes cross-domain retrieval, leveraging a diverse dataset from three countries with geo-tags. It employs the Large Multimodal Model (LMM) for textual refinement and the Segment Anything Model (SAM) for visual augmentation, resulting in fine-grained alignment of images, segments, and texts. UrbanCross also incorporates an adaptive curriculum-based source sampler and a weighted adversarial cross-domain fine-tuning module, enhancing adaptability across domains.

**Strengths:**

1.UrbanCross proposes a novel cross-domain adaptation mechanism for handling data distribution disparities across different geographical locations and urban environments. By employing an adaptive curriculum-based source sampler and a weighted adversarial cross-domain fine-tuning module, UrbanCross effectively bridges the gaps between different domains.

2.UrbanCross enriches text descriptions by integrating geographical labels and leveraging a large-scale multi-modal model (LMM). Additionally, it employs Segment Anything Model (SAM) for precise visual segmentation, enhancing the model's understanding and recognition of urban features.

**Limitations:**

1. While UrbanCross demonstrates excellent cross-domain adaptability, the paper may not have fully explored the model's generalization capabilities across a broader range of geographical and cultural contexts. If testing is limited to specific countries or cities, the model's effectiveness in other regions remains an open question.

2. The paper may lack detailed discussion on the computational complexity of the UrbanCross framework, including resource consumption during training and inference.  Given the utilization of Large Multimodal Models (LMM) and Segment Anything Model (SAM), there might be high computational resource requirements, potentially limiting the model's application in resource-constrained environments. Efficiency and scalability of the model are equally crucial for practical applications.

3. The dataset used in the paper may have limitations in certain aspects, such as potentially lacking sufficient diversity or containing biases. The authors could discuss the potential biases in the dataset and the impact they may have on the model's performance.

4. For certain applications like emergency response or real-time monitoring, the model's real-time performance is crucial. The paper may not have assessed UrbanCross's performance in real-time environments.

5. While the paper thoroughly investigates hyperparameters, it may not explore automated hyperparameter automated hyperparameter optimization methods like Bayesian optimization, which could save significant time and resources in practical applications.

6. The paper may not have conducted sufficient comparisons with current state-of-the-art techniques, such as the method proposed in "Knowledge-Aided Momentum Contrastive Learning for Remote-Sensing Image Text," and also may not have thoroughly explained the rationale behind selecting specific baseline models for comparison.

7. It seems like there might be some misstatements in the paper, such as the confusion between "i" and "j" on line 417, the correction of "cross-domain" instead of "cross-modal" on line 458, inconsistency in the usage of "target batch" and "source batch" with their corresponding "nt" and "ns" in Figure 3, and discrepancies between the statement on lines 786-788 and the results in Table 2. These errors should be addressed for clarity and accuracy in the paper.

**Suitability:**

3

---

### Official Review · Reviewer_6aeG · 2024-05-24

**Rating:** 4
**Confidence:** 2

**Summary:**

This paper introduces UrbanCross, an innovative framework for cross-domain satellite image-text retrieval that addresses the diversity challenge in urban landscapes. By leveraging a multi-country dataset and advanced models such as LMM and SAM, it achieves a 10% improvement in retrieval performance. Additionally, it includes adaptive and adversarial fine-tuning mechanisms that enhance adaptability across domains, resulting in an average 15% performance increase over non-adapted versions.

**Strengths:**

1. This work introduces a novel approach to cross-domain satellite image-text retrieval. It leverages high-quality cross-domain datasets enriched with geo-tags and employs advanced models like the Large Multimodal Model (LMM) and the Segment Anything Model (SAM) for enhanced visual and textual alignment.
2. Extensive experiments demonstrate that UrbanCross achieves significant improvements in retrieval performance. It enhances retrieval efficiency by 10% and outperforms methods lacking domain adaptation mechanisms by 15% on average.
3. The paper clearly outlines its contributions, including data augmentation, cross-domain adaptation, and extensive experimentation. This makes it easy for readers to understand the key innovations and improvements introduced by UrbanCross.

**Limitations:**

1. The proposed framework is comprehensive and involves multiple sophisticated components, such as LMM, SAM, adaptive curriculum-based source sampling, and weighted adversarial fine-tuning. However, it is challenging to implement in realistic scenes.

2.The hyperparameter analysis is inadequate. The authors conducted experiments on several hyperparameters, selecting only three values for comparison. It is recommended that the analysis be expanded with additional explanations.

**Suitability:**

3

---

### Official Review · Reviewer_S44T · 2024-05-26

**Rating:** 4
**Confidence:** 3

**Summary:**

This paper introduces a novel UrbanCross framework which enhances satellite image-text retrieval by addressing domain gaps across diverse urban landscapes through cross-domain adaptation. It leverages high-quality texts, geo-tags, and LMM integration to improve image-text alignment, especially in datasets with thorough semantics and contextual information. The framework demonstrates significant adaptability improvements in domain shifts across various countries, with mean recall increases of 15.0% for Finland to Spain, 15.2% for Germany to Spain, and 15.5% for Germany to Finland when fully configured with SS, CL, and AT components. The fine-tuning stage optimizes model performance during domain adaptation by adjusting learning rates and batch sizes to maximize mean recall while ensuring model stability. Additionally, qualitative analysis showcases the efficacy of domain adaptation in improving image-text alignment, as demonstrated in examples of Finland to Spain and Germany to Finland transfers.

**Strengths:**

a.	This paper significantly enhances both visual and textual accuracy by integrating LMM with geo-tags to enrich textual descriptions and employing SAM for precise visual segmentation.
b.	This paper introduces a curriculum-based source sampler and a weighted adversarial fine-tuning module which improves domain adaptation.
c.	Extensive experimental results demonstrate the compelling performance of the method compared to the SOTA baselines.

**Limitations:**

a.	The method utilizes Vit-L-14 as the backbone, what if replace it with Vit-B-16? Since it is noted that the RemoteCLIP which utilizes Vit-B-16 also achieves good results, this paper should report the backbone results for fair comparison.
b.	This paper should report the results of other methods on the UC dataset, for a more solid comparison. And the UC dataset was also used in the ablation experiment, are the results of the ablation experiment representative of the validity on the other two datasets (i.e. RSICD and RSITMD)?
c.	In the hyperparametric experiments, it is noted that optimal performance is reached with epoch = 15 during Pre-training, as shown in Figure 5 (b), but this paper does not report performance with epoch less than 15, which may not be a good proof that 15 is optimal.

Typos and minors
It would be advisable to check the entire manuscript with a view to the typos, grammar and syntax. For example, singular and plural expressions should be checked.

**Suitability:**

3

---

### Official Review · Reviewer_z4XT · 2024-05-27

**Rating:** 4
**Confidence:** 2

**Summary:**

The paper addresses the need for effective satellite image-text retrieval methods for urban applications, highlighting the often overlooked domain gaps across diverse urban landscapes. To address this, the authors present UrbanCross, a new framework for cross-domain satellite image-text retrieval. UrbanCross utilizes a high-quality, cross-domain dataset with extensive geo-tags from three countries to emphasize domain diversity. It employs the Large Multimodal Model (LMM) for textual refinement and the Segment Anything Model (SAM) for visual augmentation, achieving a 10% improvement in retrieval performance. Additionally, it incorporates an adaptive curriculum-based source sampler and a weighted adversarial cross-domain fine-tuning module to enhance adaptability across various domains. Experiments show a 15% performance increase over its non-adaptive version, effectively bridging the domain gap.

**Strengths:**

The paper addresses the need for effective satellite image-text retrieval methods for urban applications, highlighting the often overlooked domain gaps across diverse urban landscapes. To address this, the authors present UrbanCross, a new framework for cross-domain satellite image-text retrieval. UrbanCross utilizes a high-quality, cross-domain dataset with extensive geo-tags from three countries to emphasize domain diversity. It employs the Large Multimodal Model (LMM) for textual refinement and the Segment Anything Model (SAM) for visual augmentation, achieving a 10% improvement in retrieval performance. Additionally, it incorporates an adaptive curriculum-based source sampler and a weighted adversarial cross-domain fine-tuning module to enhance adaptability across various domains. Experiments show a 15% performance increase over its non-adaptive version, effectively bridging the domain gap.

**Limitations:**

1.	In Section 1, it is recommended to add “:”at the end of this sentence “In this paper, we identify two key perspectives to enhance satellite image-text retrieval with cross-domain adaptation”.
2.	Please check the following part. In Section 3.11, it is suggested to remove the n in the last sentence.
3.	It is recommended to write “where” at the beginning of a new line when describing the meaning of symbols in formula (2)(3)(4)(7).
4.	Each symbol in the paper should be explained as much as possible, such as what n represents in Equation (4).
5.	In Section 3.2.3, the author used a subscript for ET. Should EI and ES also use subscripts? Please check the related part
6.	If there is an overall loss function, it would be better to include it.
7.	Mean Recall has been defined as MeanR in experimental section, it is better to denote 	Mean Recall with MeanR in performance tables.

**Suitability:**

3

---

### Meta-Review · Area_Chair_W4ii · 2024-07-03

**Recommendation:** Accept (Poster)
**Confidence:** 4

**Metareview:**

This paper introduces a novel framework called UrbanCross, designed for satellite image-text retrieval tasks. It addresses domain gaps across diverse urban landscapes through cross-domain adaptation. All reviewers have acknowledged the effectiveness of the proposed framework and its innovative integration of LLM and SAM  to enhance the image-text alignment in the training data. Common concerns were initially raised regarding the model’s complexity and the evaluation of hyperparameters. However, most of these issues have been satisfactorily addressed after the rebuttal. Given its novelty and impressive performance, I recommend accepting this paper.